# The State of Trace Elements (In, Cu, Ag) in Sphalerite Studied by X-Ray Absorption Spectroscopy of Synthetic Minerals

**Nikolay D. Trofimov [1], Alexander L. Trigub [2,\*], Boris R. Tagirov [1], Olga N. Filimonova [1], Polina V. Evstigneeva [1], Dmitriy A. Chareev [3], Kristina O. Kvashnina [4,5,6] and Maximilian S. Nickolsky [1]**

[1] Institute of Geology of Ore Deposits, Petrography, Mineralogy and Geochemistry (IGEM RAS), 35 Staromonetnyi per., 119017 Moscow, Russia; trofim-kol@mail.ru (N.D.T.); boris1t@yandex.ru (B.R.T.); oliel@list.ru (O.N.F.); evstpolinav@gmail.com (P.V.E.); mnickolsky@gmail.com (M.S.N.)

[2] National Research Centre 'Kurchatov Institute', 1 Akademika Kurchatova pl., 123182 Moscow, Russia

[3] Institute of Experimental Mineralogy (IEM RAS), 142432 Chernogolovka, 142432 Moscow Region, Russia; charlic@mail.ru

[4] The Rossendorf Beamline at ESRF—The European Synchrotron, CS40220, 38043 Grenoble CEDEX 9, France; kristina.kvashnina@esrf.fr

[5] Helmholtz-Zentrum Dresden-Rossendorf (HZDR), Institute of Resource Ecology, P.O. Box 510119, 01314 Dresden, Germany

[6] Department of Chemistry, Lomonosov Moscow State University, 119991 Moscow, Russia

\* Correspondence: alexander.trigub@gmail.com

**Abstract:** The oxidation state and local atomic environment of admixtures of In, Cu, and Ag in synthetic sphalerite crystals were determined by X-ray absorption spectroscopy (XAS). The sphalerite crystals doped with In, Cu, Ag, In–Cu, and In–Ag were synthesized utilizing gas transport, salt flux, and dry synthesis techniques. Oxidation states of dopants were determined using X-ray absorption near edge structure (XANES) technique. The local atomic structure was studied by X-ray absorption fine structure spectroscopy (EXAFS). The spectra were recorded at Zn, In, Ag, and Cu $K$-edges. In all studied samples, In was in the 3+ oxidation state and replaced Zn in the structure of sphalerite, which occurs with the expansion of the nearest coordination shells due to the large In ionic radius. In the presence of In, the oxidation state of Cu and Ag is 1+, and both metals can form an isomorphous solid solution where they substitute for Zn according to the coupled substitution scheme $2Zn^{2+} \leftrightarrow Me^+ + In^{3+}$. Moreover, Ag $K$-edges EXAFS spectra fitting, combined with the results obtained for In- and Au-bearing sphalerite shows that the Me-S distances in the first coordination shell in the solid solution state are correlated with the ionic radii and increase in the order of Cu < Ag < Au. The distortion of the atomic structure increases in the same order. The distant (second and third) coordination shells of Cu and Ag in sphalerite are split into two subshells, and the splitting is more pronounced for Ag. Analysis of the EXAFS spectra, coupled with the results of DFT (Density Function Theory) simulations, showed that the In–In and $Me^+$–$In^{3+}$ clustering is absent when the metals are present in the sphalerite solid solution. Therefore, all studied admixtures (In, Cu, Ag), as well as Au, are randomly distributed in the matrix of sphalerite, where the concentration of the elements in the "invisible" form can reach a few tens wt.%.

**Keywords:** sphalerite; synthetic minerals; X-ray absorption spectroscopy; XANES; EXAFS; indium; copper; silver; trace elements

## 1. Introduction

　　Sphalerite ZnS is a unique mineral which can reach high concentrations of elements that are in high demand in the hi-tech industry. These admixtures include "critical" metals In, Cd, Ga, and Ge. Indium, being a rare element in the Earth's crust, is not concentrated enough to form its ore minerals, but it can be recovered as a byproduct during the refinement of Zn ores, where it is contained in sphalerite. X-ray spectroscopy studies of In's charge state and its local atomic structure were presented in our previous study [1]. In natural sphalerites, the concentration of In is directly correlated to the concentration of Cu [2,3]. Therefore, it is generally accepted that the formation of In-bearing sphalerite takes place via the coupled substitution scheme $2Zn^{2+} \leftrightarrow Cu^+ + In^{3+}$. The phase relations in the system $ZnS–CuInS_2$ were studied by Schorr et al. [4,5] (and references therein). It was determined that Cu and In form a partial binary solid solution, $Zn_{2x}Cu_{1-x}In_{1-x}S_2$, with a miscibility gap in the region 01; ≤, x, ≤, 04. These data confirm the $2Zn \leftrightarrow (Cu + In)$ substitution scheme. In natural ores, however, not only Cu but also other 11th group metals can coexist with In in sphalerite. In-bearing sphalerite can contain a few hundred ppm Ag (Xu et al. [6]) and a few ppm Au [2], together with other minor and trace elements which can affect the state of In.

　　This study aims to investigate the substitution mechanisms in In-bearing sphalerite by means of X-ray absorption spectroscopy (XAS). Due to low concentrations (<100 ppm in most cases), the form of occurrence of In in natural sphalerite is difficult to determine by spectroscopic methods. Therefore, for the present study, we synthesized sphalerite crystals in the systems Zn–In–S, Zn–Cu–S, Zn–Ag–S, Zn–In–Cu–S, and Zn–In–Ag–S and characterized the state of In, Cu, and Ag by XAS, with an emphasis on the extended X-ray absorption fine structure (EXAFS) technique. These measurements make it possible to determine the substitution mechanism, describe the state of the admixtures (the valence state and the local atomic environment), characterize the effect of the 11th group metal type and concentration on the state of In and, taking into account our recent study of In–Au-bearing sphalerites [1], to identify the systematic changes in the states of the 11th group metals owing to the increase in the ionic radii.

## 2. Materials and Methods

### 2.1. Synthesis

　　In the present study, we synthesized sphalerite samples doped with one or two elements. The first group includes samples doped with In (Sample 3757), Cu (Sample 4065), and Ag (Sample 4152). The second group consists of sphalerites doped with In + Cu (Samples 4108, 4186) and In + Ag (Samples 4169, 4197). The reference materials included pure sphalerite ZnS, roquesite ($CuInS_2$), and laforetite ($AgInS_2$). The initial composition of the samples is given in the second column of Table 1. The crystal growth experiments were performed using (i) the gas transport method, (ii) the salt flux technique (KCl/NaCl eutectic mixture: Chareev et al. [7,8]), and (iii) the dry synthesis method [9]. In the gas transport method, $NH_4Cl$ and $I_2$ were used as transport agents. The initial phases ~0.5 g of ZnS (synthetic sphalerite), several milligrams of $Ag_2S$ and $Cu_2S$, synthesized in our laboratory, and commercial $In_2S_3$ were powdered and loaded into a silica glass ampoule (8 mm ID, 11 mm OD, ~110 mm length). The appropriate synthesis method for each system was chosen according to Chareev et al. [10], where the sphalerite synthesis techniques are described in detail.

　　Sample 3757 (starting composition ZnS + 0.3 mol.% $In_2S_3$) was prepared using the gas transport method. $NH_4Cl$ was used as a transport agent. Ampoules filled with starting reagents were evacuated, sealed, and placed into a horizontal tube furnace with a steady-state temperature gradient. The furnace was heated to the synthesis temperature for 2–3 h and then kept at this temperature for 20 days. The temperature gradient in the furnace was 50–100 °C; the temperature at the hot end of the ampoules was 850 °C. At the end of the experiment, the ampoules were quenched in cold water.

**Table 1.** Chemical composition of synthesized sphalerite crystals.

| Sample No. | Dopant | Starting Materials, mol.% of Dopant | Starting Materials, wt.% of Dopant | Phase Composition (SEM, XRD) | EPMA, wt.% [d] | | | | | LA-ICP-MS, ppm [d] | |
|---|---|---|---|---|---|---|---|---|---|---|---|
| | | | | | Zn | S | Cu | Ag | In | Cu | Ag |
| 3757 [a] | In | 0.30 mol.% In$_2$S$_3$ | 0.70 wt.% In | ZnS | | | - | - | - | - | - |
| 4108 [b] | Cu + In | 0.07 mol.%Cu$_2$S 0.04 mol.% In$_2$S$_3$ | 0.10 wt.%Cu 0.09 wt.% In | ZnS | 66.26 ± 0.52 | 32.84 ± 0.23 | bdl | - | 0.12 ± 0.02 | 890 ± 3 0 | - |
| 4186 [c] | | 2.21 mol.% Cu$_2$S .21 mol.% In$_2$S$_3$ | 2.71 wt.% Cu 4.90 wt.% In | ZnS | - | - | - | - | - | - | - |
| 4065 [a] | Cu | 0.60 mol.% Cu$_2$S | 0.78 wt.% Cu | ZnS | 66.24 ± 0.40 | 32.86 ± 0.50 | 0.28 ± 0.05 | - | - | 2380 ± 90 | - |
| 4197 [c] | Ag + In | 2.5 mol.% Ag$_2$S 2.5 mol.% In$_2$S$_3$ | 5.04 wt.% Ag 5.37 wt.% In | ZnS + Ag$_2$S | 58.37 ± 2.28 | 31.95 ± 1.04 | - | 4.74 ± 0.82 | 5.15 ± 0.46 | - | - |
| 4169 [a] | | 0.05 mol.% Ag$_2$S 0.04mol.% In$_2$S$_3$ | 0.11 wt.% Ag 0.09 wt.%In | ZnS | 68.22 ± 1.68 | 33.94 ± 0.92 | - | 0.01 ± 0.06 | 0.07 ± 0.08 | - | 180 ± 7 |
| 4152 [a] | Ag | 0.40 mol.% Ag$_2$S | 0.90 wt.% Ag | ZnS | 66.49 ± 0.50 | 33.40 ± 0.72 | - | 0.08 ± 0.08 | - | - | 23 ± 1 |

[a] gas transport method, synthesis at 850/750 °C (hot/cold end of the ampoule); [b] salt flux method (KCl/NaCl eutectic mixture), synthesis at 790/730 °C (hot/cold end of the ampoule); [c] dry synthesis method synthesis at 550 °C; [d] bdl—below the limit of detection; dash—not measured; for Samples 3757 and 4186, the amount of synthesis products was insufficient to perform chemical analyses by means of EPMA and LA-ICP-MS; only XRD data are available for these samples.

Samples 4065 (starting composition ZnS + 0.6 mol.% $Cu_2S$) and 4108 (starting composition ZnS + 0.07 mol.% $Cu_2S$ + 0.04 mol.% $In_2S_3$) were prepared using the salt flux method in the steady-state temperature gradient. The ampoules were prepared as described above. The temperature was set to 790 °C (hot end) and 734 °C (cold end).

Samples 4152 (starting composition ZnS + 0.4 mol.% $Ag_2S$) and 4169 (starting composition ZnS + 0.05 mol.% $Ag_2S$ + 0.04 mol.% $In_2S_3$) were synthesized using the gas transport method, with iodine as a transport reagent. The temperature was set to 850 °C (hot end) and 750 °C (cold end).

Samples 4197 (starting composition ZnS + 2.5 mol.% $Ag_2S$ + 2.5 mol.% $In_2S_3$) and 4186 (starting composition ZnS + 2.21 mol.% $Cu_2S$ + 2.21 mol.% $In_2S_3$) were prepared by the dry synthesis method in silica glass ampoules filled with powdered sulfides. The synthesis was performed at 550 °C for 28 days. After two weeks, the ampoules were quenched, and the samples were finely ground to provide higher homogeneity, sealed again in the ampoules, and kept at the synthesis temperature for the second time during the last two weeks. The reference materials were prepared by the dry synthesis method.

## 2.2. Analytical Methods

The morphology of the synthesized minerals was studied using scanning electron microscopy (SEM), phase compositions were characterized using X-ray diffraction (XRD), and chemical composition was determined by means of electron probe micro-analysis (EPMA) and laser ablation inductively coupled mass spectrometry (LA-ICP-MS). The EPMA and LA-ICP-MS analyses were performed on polished sphalerite grains mounted in epoxy. The SEM studies were performed using a JSM-5610LV electron microscope (JEOL LDT, Akishima, Japan) equipped with an X-Max-80 energy-dispersive spectrometer (Oxford Instruments, Oxford, UK). EPMA analyses were performed using a JEOL JXA-8200 WD/ED (JEOL LDT, Akishima, Japan) combined electron probe microanalyser (JEOL LDT, Akishima, Japan) equipped with five wavelength-dispersive X-ray spectrometers (JEOL LDT, Akishima, Japan). The operating conditions were as follows: 20 kV accelerating voltage, 20 nA beam current, and counting time of 10 s. The lines and diffracting crystals used were as follows: Zn and Cu, Kα (LiF crystal); In, Lα (PETH); Ag, Lα (PET); S, Kα (PETH). Calibration reference materials were as follows: pure sphalerite ZnS (for Zn), InSb (for In), and $Ag_2S$ (for Ag). The limits of detection (2σ, wt.%) were 0.07 (Zn), 0.06 (Cu), 0.03 (In), and 0.03 (Ag). Concentrations of $^{63}$Cu and $^{107}$Ag isotopes in the synthesized sphalerite crystals, and the distribution modes (homogeneous/inhomogeneous) of Cu and Ag were determined using a New Wave 213 laser (ESi) (ESI, Omaha, NE, USA) coupled with the Thermo Scientific X Series 2 quadrupole ICP-MS (ESI, Omaha, NE, USA)). The laser pulse frequency was 10 Hz with a power of 6–8 J/cm$^2$ and the beam size of 40–60 μm. The analysis was carried out for 30 s, followed by 20 s for the gas blank. The ablation was performed in He + 6% $H_2$ (0.6 L/min) atmosphere. The gas carrying ablated material to the ICP mass spectrometer was mixed with Ar (0.8 L/min). Sulfide reference material MASS-1 [11] was used as an external calibration standard for both In and Cu, together with in-house pyrrhotite $Fe_{0.9}S$. Isotopes $^{68}$Zn or $^{33}$S were used as internal standards.

X-ray diffraction analysis was performed on synchrotron radiation-based (λ = 0.8 Å) XRD data in the 2θ range, from 4° to 45°, using a Rayonix SX 165 detector. All measurements were taken at the Synchrotron Center of the National Research Center, "Kurchatov Institute". LaB$_6$ powder (NIST SRM 660a) (NIST, Geithersburg, MD, USA) was used as standard.

## 2.3. X-Ray Absorption Spectroscopy (XAS) Measurements

X-ray absorption experiments (i.e., collection of X-ray absorption spectra in the energy range near the core-level of the specified element) were performed at the European Synchrotron Radiation Facility (ESRF) in Grenoble, France. An X-ray absorption spectrum comprises XANES (X-ray absorption near edge structure) data (~50–80 eV from the absorption edge), which provides information about the electronic structure and coordination geometry, and EXAFS data (~1000 eV from the absorption edge),

which provides information on the radii of coordination shells and coordination numbers around the absorbing atom [12]. The In, Ag, and Cu *K*-edge spectra were recorded at the Rossendorf Beamline BM20 of the ESRF. The storage-ring operating conditions were 6.0 GeV and 80–100 mA. The photon energy was scanned from 27,700 to 28,570 eV (In), 8600 to 9900 eV (Cu), and 25,300 to 26,250 eV (Ag), using the Si (111) monochromator coupled to Rh-coated mirrors for the collimation and reduction of higher harmonics. Energy calibration was performed using the *K*-edge excitation energy of Zn (9659 eV), In (27,940 eV), Cu (8979 eV), and Ag (25,514 eV) metal foil. The spectra of the metal foils, placed after the samples and between the second and third ionization cameras, were recorded in transmission mode simultaneously with the spectra of the samples. The spectra of reference substances were collected in transmission mode, while the spectra of sphalerite samples were recorded in total fluorescence yield mode using a 13-element high-throughput Ge-detector. The detected intensity was normalized to the incident photon flux.

### 2.4. XANES Spectra Analysis

The normalized XANES spectra were analyzed by determining the characteristics of the main spectral features (absorption edge and white line positions) using the ATHENA program (IFEFFIT software package (version 08.056, NIST, Geithersburg, MD, USA) [13]). It is well known that XANES spectra are sensitive not only to the oxidation state of absorbing atoms but also to the local atomic environment. The most straightforward approach to XANES data processing is a linear combination fitting (LCF) [14]. In the LCF method, the X-ray absorption spectrum is approximated as a linear combination of the spectra of standards. The ATHENA program was used for the LCF analysis.

### 2.5. EXAFS Spectra Fitting

The EXAFS ($\chi_{exp}(k)$) data were analyzed using the ARTEMIS program (a part of IFEFFIT software package) (version 08.056, NIST, Geithersburg, MD, USA). Following standard procedures for pre-edge subtraction and background removal, the structural parameters—interatomic distances ($R_i$), coordination numbers ($N_i$), and Debye–Waller factors ($\sigma^2_i$)—were determined via the non-linear fit of theoretical spectra to the experimental ones with the equation

$$\chi(k) = S_0^2 \sum_{i=1}^{n} \frac{N_i F_i(k)}{R_i^2 k} e^{\frac{-2R_i}{\lambda(k)}} e^{-2\sigma_i^2 k^2} \sin(2kR_i + \varphi_i(k)) \tag{1}$$

Theoretical spectra were simulated using photoelectron mean free path length $\lambda(k)$, amplitude $F_i(k)$, and phase shift $\varphi_i(k)$ parameters, which were calculated ab initio using the program FEFF6 [15]. For the FEFF6 calculation, the crystal structure of sphalerite was used as a model of the solid solution. The Wavelet transform (WT) analysis of the EXAFS spectra was applied in order to discriminate contributions of heavy atoms to the EXAFS signal and clarify the differences in the local atomic structure between samples and reference substances. Details of the WT analysis are given in the Supplementary Materials.

### 2.6. DFT Calculations

In our previous studies [1,16,17], good precission of DFT calculated structures was demonstrated by the comparison of bond lengths with the results of X-ray absorption spectroscopy. In the present study, the DFT technique was used to estimate the parameters of locala atomic structure around impurity atoms in sphalerite. The software package QUANTUM ESPRESSO [18] was used for the calculations. We employed a projector-augmented wave description of the electron–ion interactions [19,20] with the Perdew–Burke–Ernzerhof (PBE) exchange-correlation functional. The self-consistent field (SCF) method, with a 70 Ry kinetic energy cut-off for the plane waves, a 1000 Ry charge density cut-off, and a SCF tolerance of $10^{-9}$ Ry, was applied in the electronic structure calculations. The optimizations of the crystal structure and supercell parameters were performed using the Broyden–Fletcher–Goldfarb–Shanno

algorithm for the atomic coordinates, with a convergence threshold of $10^{-3}$ Ry/au for the forces and of $10^{-4}$ Ry for the energy. The relaxation of the atomic positions and cell parameters was applied for a $3 \times 3 \times 3$ supercell, which contained 108 Zn (or dopants and Zn atoms) and 108 S atoms, with periodic boundary conditions. In all cases, the large unit cell allowed the gamma point approximation to be employed.

## 3. Results

### 3.1. Phase and Chemical Composition of Samples

The crystals obtained in the present study are similar in morphology, size, and character of the distribution of admixtures to those described in our previous studies [1,21]. Examples of the synthesized phases in the system Zn–In–Ag–S are shown in Figure 1. The phase composition of samples and the chemical composition of sphalerite crystals are listed in Table 1.

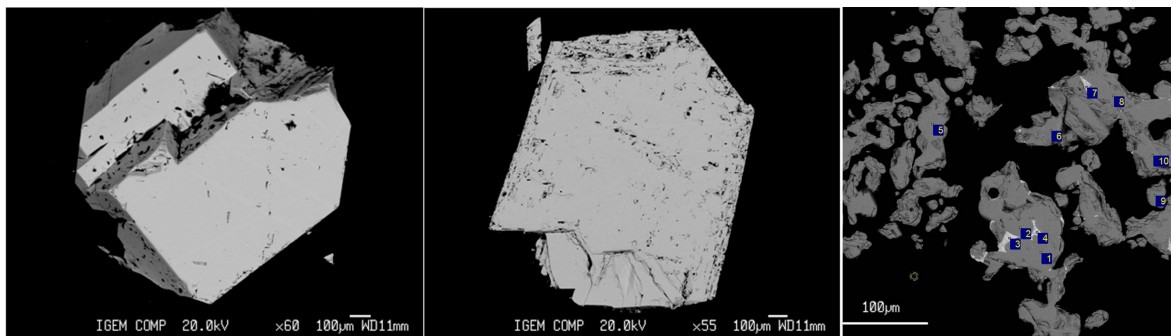

**Figure 1.** Back-scattered electron images of sphalerite crystals synthesized in the Zn–Ag–(In)–S system. Samples 4152, 4169—gas transport synthesis, Sample 4197—dry synthesis. Several inclusions of Ag$_2$S (light-grey) are seen in the image of Sample 4197; numbers indicate points of analysis.

The XRD patterns of all the samples corresponded to pure sphalerite PDF#5-566 within the measurement error. No other phases were detected in the studied samples by means of XRD. As demonstrated in our previous study [21], In is homogeneously distributed in sphalerite crystals synthesized in the system Zn–In–S at 850 °C at In concentrations up to at least 0.5 wt.%.

The admixtures in Samples 4065 and 4108 (Zn–In–Cu system) are distributed homogeneously, which is proven by the small variation in measured concentrations (see uncertainties of measured concentrations in Table 1). The SEM/EDS analyses revealed that Sample 4197 (Zn–In–Ag–S system) contains two phases (Figure 1, right panel). The EPMA analysis of Sample 4197 detected the presence of traces of Ag$_2$S, along with Ag-bearing sphalerite. Note that, despite high concentrations of doping elements Ag and In, both elements present in the "invisible" state in sphalerite of the Sample 4197.

### 3.2. XANES Spectra Analysis

Figure 2a,b show the coordination polyhedra around a cation in the sphalerite and roquesite CuInS$_2$ structures according to data from the literature. The interatomic distances from the central cation to the vertexes of the first, second, and third coordination shells in the sphalerite, roquesite, and laforetite AgInS$_2$ structures are given in Table 2. The In, Cu, and Ag $K$-edge XANES spectra of sphalerite samples and reference substances are shown in Figure 3. Energy positions of the absorption edge (e.j.) and the first intense spectral feature, white line (WL), are listed in Table S1 (Supplementary Materials).

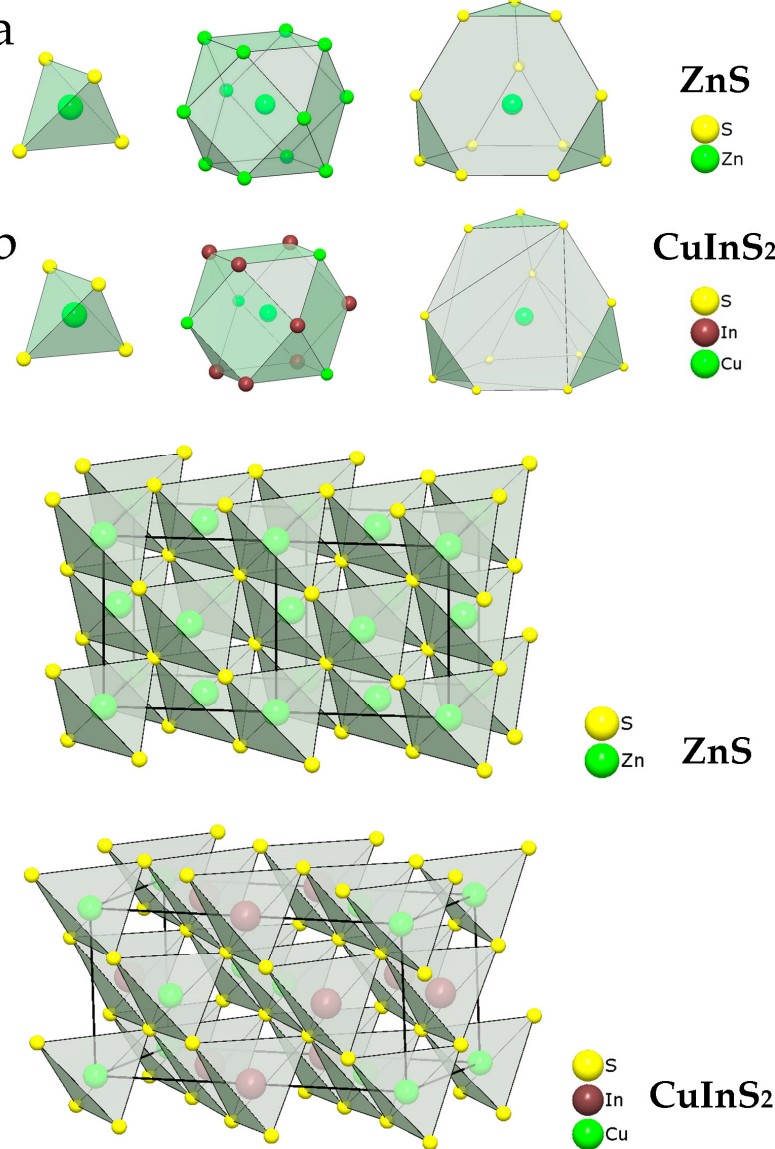

**Figure 2. Top:** First, second, and third coordination polyhedra around a cation in the structures of sphalerite ZnS (**a**) [22] and roquesite $CuInS_2$ (**b**) [23]. **Bottom:** Crystal structures of sphalerite (doubled cell is shown to make comparison with $CuInS_2$ easier) and roquesite [23].

**Table 2.** Interatomic distances in the three nearest-to-cation coordination shells (CS) in sphalerite ZnS, roquesite $CuInS_2$, and laforetite $AgInS_2$ according to crystal chemical data.

| Formula | Mineral | Space Group | Central Cation | 1st CS (Å) | 2nd CS (Å) | 3rd CS (Å) |
|---------|---------|-------------|----------------|-----------|-----------|-----------|
| ZnS [a] | Sphalerite | F-43m | Zn | 4S: 2.3427 | 12Zn: 3.8256 | 12S: 4.4859 |
| $CuInS_2$ [b] | Roquesite | I-42d | Cu | 4S: 2.3287 | 4In: 3.9039<br>4In: 3.9204<br>4Cu: 3.9204 | 4S: 4.5457<br>4S: 4.6477<br>4S: 4.6929 |
| $AgInS_2$ [c] | Laforetite | I-42d | Ag | 4S: 2.5543 | 4Ag: 4.0587<br>4In: 4.0587<br>4In: 4.1550 | 4S: 4.6608<br>4S: 4.7772<br>4S: 4.8773 |

[a] ZnS: Jamieson and Demarest [22]; [b] $CuInS_2$: Abrahams et al. [23]; [c] $AgInS_2$: Delgado et al. [24].

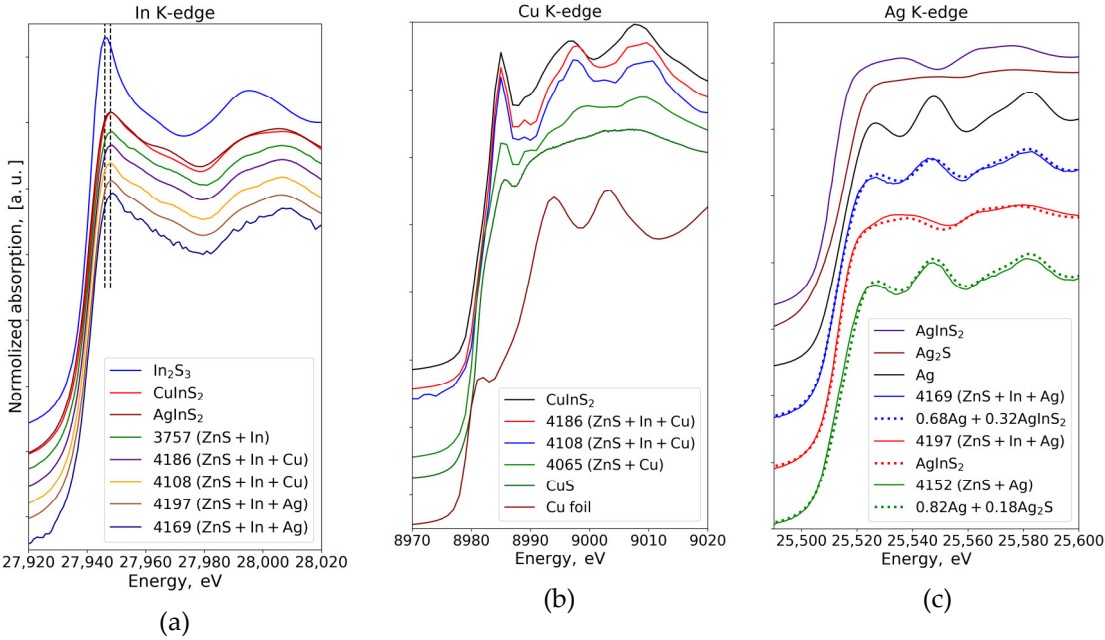

**Figure 3.** X-ray absorption near edge structure (XANES) spectra of synthetic sphalerites and reference substances: (**a**) In *K*-edge spectra; (**b**) Cu *K*-edge spectra; (**c**) Ag *K*-edge spectra. The qualitative composition of the sphalerite samples and the references are indicated in each figure. Dashed lines in (**a**) indicate the position of white line (WL) for $In_2S_3$ (left line) and all other samples and references (right line). Dotted lines in (**c**) show the results of the linear combination fitting (LCF) of the spectra of (In)–Ag–bearing sphalerites; fit results are given in Table S2 of Supplementary Materials and discussed in Section 3.2.3.

### 3.2.1. In *K*-Edge

The shapes of the spectra and positions of the spectral features (e.j. and WL) are very close for all In-bearing samples and reference substances apart from $In_2S_3$. This means that the presence of Cu and Ag, and the difference in the concentrations of these metals, has a negligible effect on the local atomic environment and the valence state of In. The "formal" oxidation state of In in sphalerite is 3+. The difference of the spectrum of Sample 3757 (In is the only dopant) from the spectrum of $In_2S_3$ suggests a solid solution as a possible form of occurrence of In in the sample. The nearest-to-In atomic geometry in the samples of In–Cu and In–Ag-bearing sphalerites is similar to roquesite $CuInS_2$ and laforetite $AgInS_2$ because of the similarity of the spectra. The coordination shells of In in $CuInS_2$ and $AgInS_2$ consist of 4 S atoms (first shell), 12 metal atoms—(In, Cu) or (In, Ag) (second shell), and 12 S atoms (third shell, Figure 2, Table 2). The composition and type (topology) of coordination polyhedra around a cation in roquesite and laforetite are identical to those of sphalerite (top images in Figure 2). The close shapes of the XANES spectra of In, In–Cu, and In–Ag-bearing sphalerites and references $CuInS_2$ and $AgInS_2$, together with the similarity of the local atomic structures of sphalerite, roquesite, and laforetite, imply that In in sphalerite exists in the form of the solid solution, where it occupies the cationic position.

### 3.2.2. Cu *K*-Edge

The spectra of sphalerite samples differ considerably from the spectrum of Cu foil. The absence of the pre-edge feature, which is characteristic of $Cu^{2+}$ electronic structure (Ar) $3d^9$, implies 1+ oxidation state of Cu in all samples [25]. The spectra of Samples 4186 and 4108, which contain two admixtures, Cu and In, are similar to the spectrum of $CuInS_2$. The spectrum of Sample 4065 (Cu-bearing sphalerite without In) differs from that of $CuInS_2$, but it is very close to the spectrum of covellite CuS. To summarize,

the local atomic geometry of Cu in In–Cu-bearing sphalerites is very close to $CuInS_2$, but in the absence of In, the Cu atoms occur mostly in the form of CuS.

### 3.2.3. Ag *K*-Edge

The shape of XANES spectra measured at the Ag *K*-edge indicates the presence of two forms of silver: $Ag°$ (the dominant form in Samples 4152 and 4169) and $Ag^+$ (Sample 4197). The systematic shift in the position of the white line is observed in the following series of samples and standards: $WL_{AgInS2} < WL_{4197} < WL_{4169} < WL_{4152} < WL_{Ag}$. This dependence is associated with an increase in the concentration of metallic $Ag°$ in this series. The application of the LCF analysis yielded an excellent description of the experimental spectra using three components: $AgInS_2$, $Ag_2S$, and metallic Ag. The comparison of the fit results (Table S2 and Figure S1 of Supplementary Materials) with the experimental spectra is given in Figure 3c. Note that the spectrum of Sample 4197, where Ag is in the oxidation state 1+, slightly differs from the spectrum of the $AgInS_2$ standard. This fact can be explained either by the presence of admixture of $Ag_2S$ detected by SEM analysis along with laforetite $AgInS_2$ or by the formation of the solid solution with an Ag local atomic structure close to $AgInS_2$.

### 3.3. EXAFS Analysis

The results of EXAFS spectra fitting for all samples and standards are collected in Table 3. Figure 4 presents the results of the wavelet transform of the EXAFS spectra. The experimental spectra are compared with the fit results in Figure 5 (reference substances, Ag, In, Cu, In *K*-edge spectra), Figure 6 (In *K*-edge), Figure 7 (Cu *K*-edge), and Figure 8 (Ag *K*-edge).

**Table 3.** Structural parameters derived from the EXAFS analysis.

| Atomic Shell | Experimental Data | | | | |
|---|---|---|---|---|---|
| | N [a,b] | R, Å | $\sigma^2$, Å$^2$ | $E_0$, eV | R-Factor |
| References | | | | | |
| Pure sphalerite ZnS, Zn *K*-edge (*k*-range = 3–14 Å$^{-1}$, R-range = 1.3–4.5 Å) | | | | | |
| S | 4 | 2.34 ± 0.01 | 0.005 ± 0.001 | | |
| Zn | 12 | 3.85 ± 0.03 | 0.017 ± 0.002 | 0.3 ± 1.3 | 0.014 |
| S | 12 | 4.46 ± 0.04 | 0.015 ± 0.004 | | |
| $CuInS_2$, In *K*-edge (*k*-range = 3–14 Å$^{-1}$, R-range = 1.3–2.5 Å) | | | | | |
| S | 4 | 2.47 ± 0.01 | 0.0026 ± 0.0005 | 5.7 ± 1.34 | 0.010 |
| $CuInS_2$, Cu *K*-edge (*k*-range=3–12 Å$^{-1}$, R-range = 1.2–2.2 Å) | | | | | |
| S | 4 | 2.31 ± 0.01 | 0.008 ± 0.001 | 4.6 ± 1.2 | 0.003 |
| $AgInS_2$, In *K*-edge (*k*-range = 3–14 Å$^{-1}$, R-range = 1.3–2.5 Å) | | | | | |
| S | 4 | 2.47 ± 0.006 | 0.0029 ± 0.0004 | 6.5 ± 1.02 | 0.012 |
| $AgInS_2$, Ag *K*-edge (*k*-range = 3–14 Å$^{-1}$, R-range = 1.3–2.5 Å) | | | | | |
| S | 4 | 2.54 ± 0.01 | 0.011 ± 0.0007 | 3.5 ± 1.2 | 0.009 |
| In *K*-edge ($S_0^2$ = 0.95) | | | | | |
| Sample 4108 (*k*-range = 3–13 Å$^{-1}$, R-range = 1.3–4.5 Å) | | | | | |
| S | 4 | 2.46 ± 0.01 | 0.003 ± 0.0002 | | |
| Zn | 12 | 3.91 ± 0.02 | 0.015 ± 0.001 | 6.3 ± 0.8 | 0.010 |
| S | 12 | 4.49 ± 0.02 | 0.011 ± 0.002 | | |
| Sample 4186 (*k*-range = 3–12 Å$^{-1}$, R-range = 1.3–4.5 Å) | | | | | |
| S | 4 | 2.46 ± 0.01 | 0.004 ± 0.0004 | | |
| Zn | 12 | 3.91 ± 0.02 | 0.016 ± 0.002 | 5.8 ± 0.9 | 0.017 |
| S | 12 | 4.47 ± 0.02 | 0.012 ± 0.003 | | |

**Table 3.** *Cont.*

| Atomic Shell | Experimental Data | | | | |
|---|---|---|---|---|---|
| | N [a,b] | R, Å | $\sigma^2$, Å$^2$ | E$_0$, eV | R-Factor |
| *Sample 3757 (k-range = 3–13 Å$^{-1}$, R-range = 1.3–4.5 Å)* | | | | | |
| S | 4 | 2.45 ± 0.01 | 0.004 ± 0.0004 | | |
| Zn | 12 | 3.91 ± 0.01 | 0.014 ± 0.0002 | 5.4 ± 0.8 | 0.036 |
| S | 12 | 4.48 ± 0.03 | 0.014 ± 0.004 | | |
| *Sample 4169 (k-range = 3–13 Å$^{-1}$, R-range = 1.3–4.5 Å)* | | | | | |
| S | 4 | 2.49 ± 0.01 | 0.004 ± 0.001 | | |
| Zn | 12 | 3.90 ± 0.02 | 0.011 ± 0.002 | 7.9 ± 1.5 | 0.050 |
| S | 12 | 4.54 ± 0.03 | 0.006 ± 0.002 | | |
| *Sample 4197 (k-range = 3–13 Å$^{-1}$, R-range = 1.3–4.5 Å)* | | | | | |
| S | 4 | 2.45 ± 0.003 | 0.004 ± 0.0002 | | |
| Zn | 12 | 3.93 ± 0.01 | 0.016 ± 0.001 | 6.4 ± 0.5 | 0.007 |
| S | 12 | 4.50 ± 0.02 | 0.016 ± 0.002 | | |
| *Cu K-edge (S$_0^2$ = 0.75)* | | | | | |
| *Sample 4186 (k-range = 3–12 Å$^{-1}$, R-range = 1.3–4.5 Å)* | | | | | |
| S | 4 | 2.31 ± 0.01 | 0.006 ± 0.001 | | |
| Zn | 5.4 ± 1.9 | 3.76 ± 0.02 | 0.011 ± 0.002 | | |
| Zn | 6.6 ± 1.9 | 3.92 ± 0.04 | 0.011 ± 0.002 | 2.2 ± 1.1 | 0.018 |
| S | 5.9 ± 2.2 | 4.31 ± 0.02 | 0.009 ± 0.004 | | |
| S | 6.1 ± 2.2 | 4.52 ± 0.02 | 0.009 ± 0.004 | | |
| *Sample 4108 (k-range = 3–11 Å$^{-1}$, R-range = 1.3–4.5 Å)* | | | | | |
| S | 4 | 2.30 ± 0.01 | 0.005 ± 0.0006 | | |
| Zn | 6.4 ± 0.9 | 3.81 ± 0.02 | 0.007 ± 0.003 | | |
| Zn | 5.6 ± 0.9 | 4.01 ± 0.04 | 0.007 ± 0.003 | 1.7 ± 1.3 | 0.01 |
| S | 12 ± 1.4 | 4.44 ± 0.02 | 0.011 ± 0.002 | | |
| *Sample 4065 (k-range = 3–12 Å$^{-1}$, R-range = 1.2–4.5 Å)* | | | | | |
| Cu_1 (D3h, trianglular geometry) | | | | | |
| S | 0.8 ± 0.2 | 2.16 ± 0.02 | 0.001 ± 0.002 | | |
| S | 2 | 3.63 ± 0.02 | 0.004 ± 0.003 | | |
| Cu_2 (Td, tetrahedral geometry) | | | | | |
| S | 1.8 ± 0.2 | 2.28 ± 0.02 | 0.003 ± 0.002 | 4.2 ± 1.1 | 0.01 |
| S | 0.6 | 2.31 ± 0.02 | 0.003 | | |
| Cu | 2 | 3.10 ± 0.03 | 0.014 ± 0.004 | | |
| S | 2 | 3.45 ± 0.02 | 0.004 ± 0.003 | | |
| *Ag K-edge (S$_0^2$ = 0.95)* | | | | | |
| *Ag foil (k-range = 3–13 Å$^{-1}$, R-range = 1.3–5.0 Å)* | | | | | |
| Ag | 12 | 2.86 ± 0.01 | 0.009 ± 0.001 | | |
| Ag | 6 | 4.02 ± 0.01 | 0.012 ± 0.001 | −0.3 ± 0.3 | 0.01 |
| Ag | 24 | 4.99 ± 0.01 | 0.014 ± 0.003 | | |
| *Sample 4152 (k-range = 3–13.5 Å$^{-1}$, R-range = 1.3–5.0 Å)* | | | | | |
| Ag | 9.4 ± 1.1 | 2.87±0.01 | 0.008 ± 0.001 | | |
| Ag | 6 | 3.95±0.07 | 0.015 ± 0.006 | 2.0 ± 0.8 | 0.02 |
| Ag | 24 | 5.00±0.02 | 0.013 ± 0.003 | | |
| *Sample 4169 (k-range = 3–14 Å$^{-1}$, R-range = 1.3–5.0 Å)* | | | | | |
| Ag | 8.6 ± 0.9 | 2.85 ± 0.01 | 0.009 ± 0.001 | | |
| Ag | 6 | 3.98 ± 0.04 | 0.018 ± 0.007 | 0.4 ± 0.6 | 0.02 |
| Ag | 24 | 4.91 ± 0.04 | 0.017 ± 0.007 | | |
| *Sample 4197 (k-range = 3–14 Å$^{-1}$, R-range = 1.3–4.5 Å)* | | | | | |
| S | 4 | 2.46 ± 0.01 | 0.008 ± 0.001 | | |
| Zn | 5.9 ± 2.9 | 3.81 ± 0.12 | | | |
| Zn | 6.1 ± 2.9 | 4.00 ± 0.23 | 0.020 ± 0.006 | 5.4±1.6 | 0.04 |
| S | 12 | 4.48 ± 0.07 | 0.025 ± 0.01 | | |

[a] Parameters without uncertainties were fixed during the fitting. [b] Sphalerite structure from [22], roquesite structure from [4] and laforetite structure from [24] were used as initial models for EXAFS fitting.

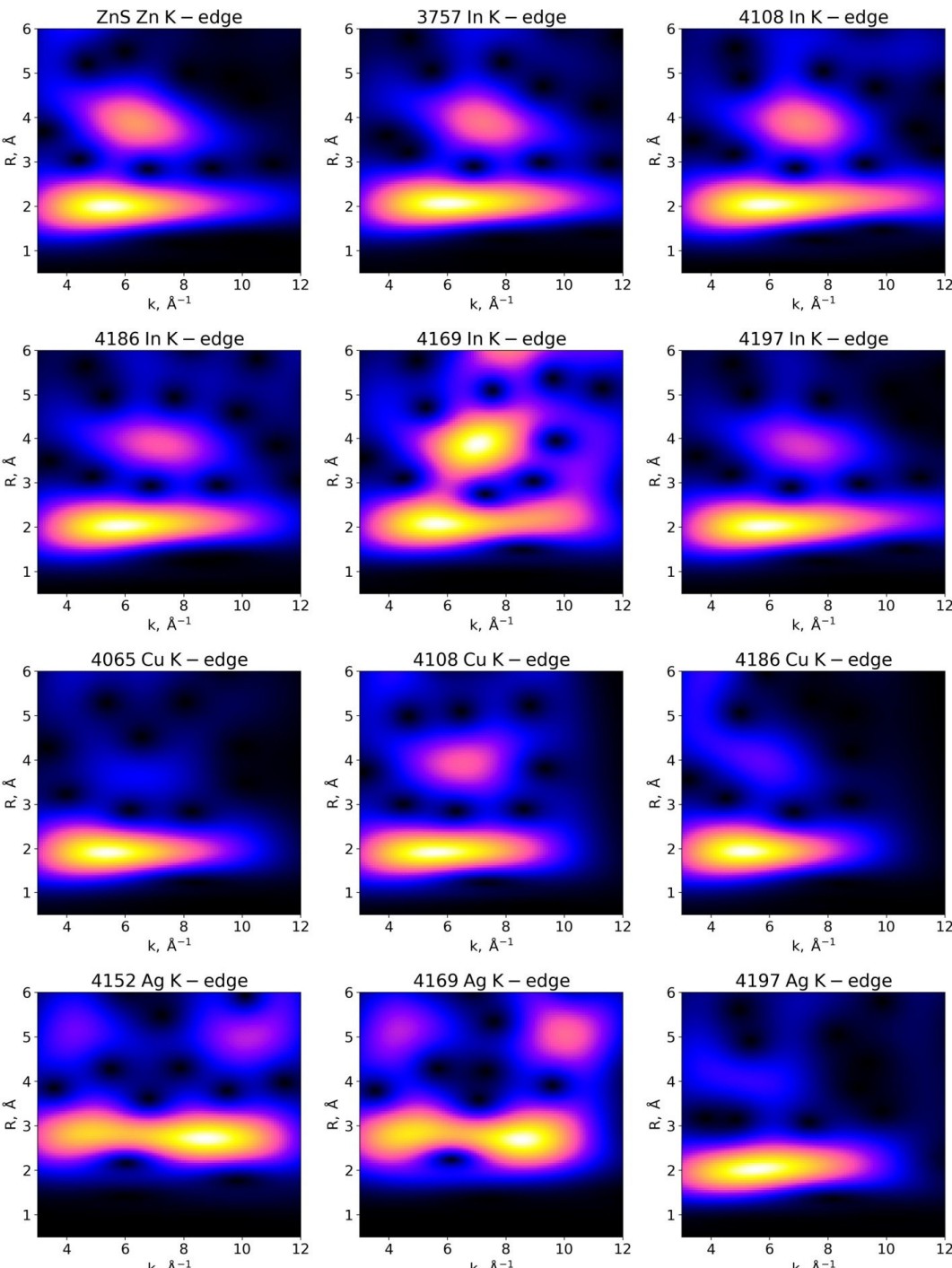

**Figure 4.** Wavelet transforms (WTs) of the X-ray absorption fine structure spectroscopy (EXAFS) signal (distance $R$ (Å) (not corrected for phase shift) vs. photoelectron wavenumber $k$ (Å$^{-1}$)). Bright colors show regions with a maximum contribution of scattering atoms located at a distance $R$ from the absorber and characterized by a definite value of the photoelectron wavenumber $k$. The similarity of the WT of ZnS and WTs obtained for all samples at In $K$-edge (Samples 3757, 4108, 4186, 4169, 4197), Cu $K$-edge (Samples 4108 and 4186), and Ag $K$-edge (Sample 4197) imply similar local atomic structures around the admixtures in these samples and that of Zn in sphalerite. The presence of a heavy atom (Ag) in the nearest coordination shell of Ag in Samples 4152 and 4169 is indicated by the contribution at $k$ ~10 Å$^{-1}$ and can be interpreted as the formation of inclusions of Ag metal.

References Ag, In, Cu, Zn K - edge spectra

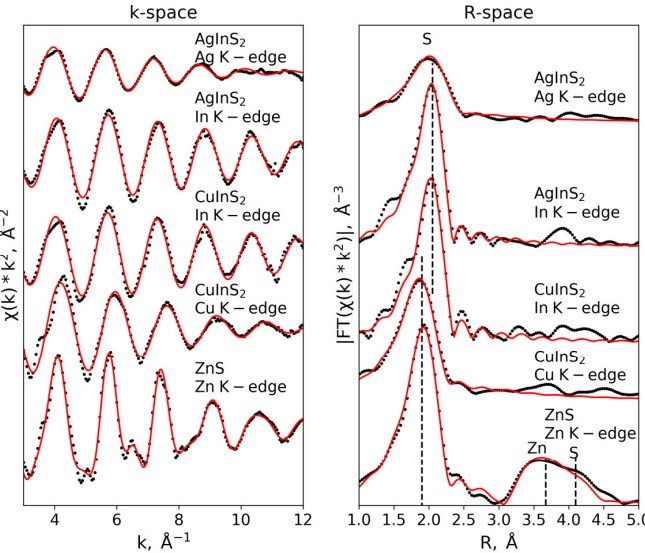

**Figure 5.** Ag, In, Cu, and Zn *K*-edge EXAFS spectra of the reference substances. Left panel: oscillation part of EXAFS spectra, k$^2$ weighted; right panel: absolute values of Fourier transforms (FTs) of EXAFS signals (not corrected for phase shift, which is usually equal to 0.3–0.5 Å). Dotted black line—experiment, solid red lines—fitted spectra (fit results are given in Table 3). Vertical lines indicate the contributions of different groups of scattering atoms. The first peak in the FT curves corresponds to four S atoms in the first coordination shell of metals. The second and third coordination shells of ZnS consist of 12 Zn and 12 S atoms, correspondingly, which contribute to the FT at a distance from 3 to 4.5 Å. The smooth character of the FTs of other references at R > 3 Å suggests distortion of the atomic structure of the distant coordination shells.

In K-edge spectra

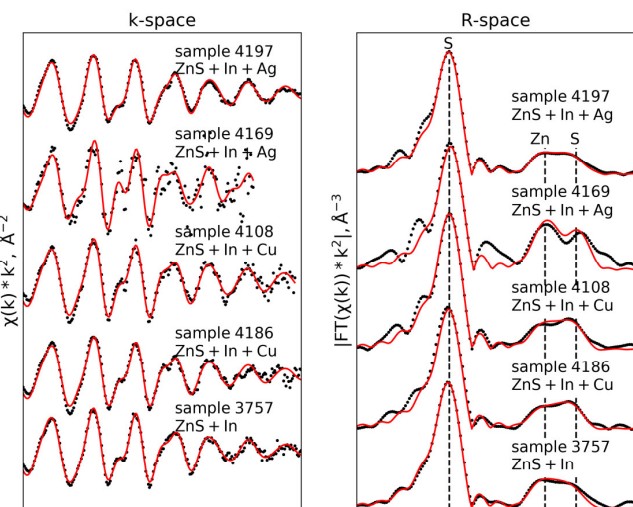

**Figure 6.** In *K*-edge EXAFS spectra of doped sphalerite samples (see caption of Figure 5 for explanation). The local atomic structure around In is not dependent on sphalerite composition and is similar to that of Zn in sphalerite. The first peak shows the presence of four S at 2.45–2.49 Å, and the peaks at 3–4.5 Å (not corrected for phase shift) correspond to 12 Zn at 3.90–3.93 Å and 12 S at 4.47–4.54 Å in the second and third coordination shells, respectively.

## Cu K-edge spectra

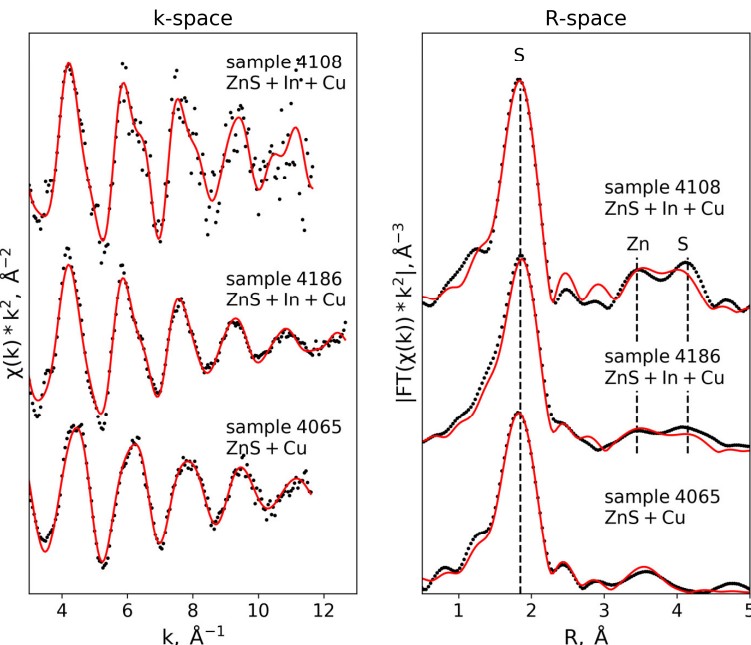

**Figure 7.** Cu *K*-edge EXAFS spectra of doped sphalerite samples (see caption of Figure 5 for explanation). In Samples 4108 and 4186, the first peak of the FT curves shows the presence of four S at 2.30–2.31 Å. The peaks at 3–4.5 Å (not corrected for phase shift) correspond to the second and third coordination shells split into two subshells. The second shell consists of (6 + 6) Zn at 3.76–4.01 Å, and the third shell consists of (6 + 6) S at 4.36–4.52 Å. The model of covellite CuS fits the spectrum of Sample 4065.

## Ag K-edge spectra

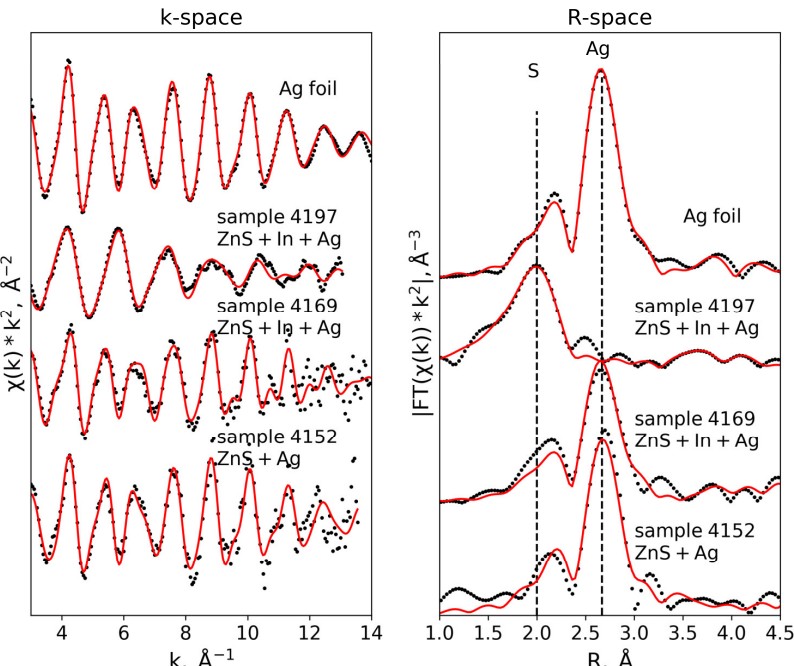

**Figure 8.** Ag *K*-edge EXAFS spectra of doped sphalerite samples and metallic Ag as a reference (see caption of Figure 5 for explanation). Sample 4197 is described as Ag-bearing sphalerite solid solution with four S at 2.46 ± 0.01 Å in the first coordination shell, 12 Zn at 3.84–4.00 Å in the second shell, and 4.48 ± 0.07 Å in the third shell. Samples 4169 and 4152 contain Ag mostly in the form of metal.

### 3.3.1. Examination of Wavelet Transforms (WTs)

Examination of the WT images (Figure 4) helps us to choose the correct structural model for the EXAFS spectra fitting and can be used to detect the heavy atoms in the nearest coordination shells around the absorbing atom [26,27]. The first image in Figure 4 (top left image) shows the WT of the Zn $K$-edge EXAFS signal of pure sphalerite. There are two main contributions of the scattering atoms to the EXAFS. The maximum of S contribution to the first sphere ($R$ ~2 Å, not corrected for phase shift) is located at the photoelectron wavenumber $k$ ~5.5 Å$^{-1}$, and the maximum of the contribution of the second sphere Zn atoms ($R$ ~3.5–4 Å) is located at $k$ ~6–6.5 Å$^{-1}$. Similar contributions of the same groups of atoms occur in the WT of the In $K$-edge EXAFS spectra of In-, In–Cu, and In–Ag-bearing sphalerites (two upper rows of Figure 4). The absence of notable heavy atom (In, Ag) contribution at high values of $k > 6.5$ Å$^{-1}$ means that the heavy atoms are uniformly distributed within the ZnS crystal structure, without the In–In and In–Ag clustering (the Cu atoms cannot be discriminated from the Zn atoms by means of EXAFS analysis). Therefore, In $K$-edge EXAFS spectra in all studied samples can be approximated by the structural model of sphalerite. In the structure of sphalerite, In substitutes for Zn, so S and Zn are the only atoms in the nearest-to-In coordination shells.

The WT of Cu $K$-edge EXAFS spectra of Samples 4108 and 4186 (third row in Figure 4) are similar to the ZnS model. In Sample 4065, the difference between the shape of the second-sphere atoms contribution and ZnS suggests nonequivalence of the local atomic environment of Cu. According to XANES spectra analysis, covellite CuS can be the main form of occurrence of Cu in Sample 4065.

The WT images of the Ag $K$-edge EXAFS signals of Samples 4152 and 4169 exhibit a pronounced peak at $k$ ~10 Å$^{-1}$, which can be attributed to the Ag–Ag bond. The WT of the EXAFS spectra of Sample 4197 is close to that of ZnS, which means that the sphalerite structure can be used to fit the spectra of the sample.

### 3.3.2. Reference Substances

Analysis of the EXAFS spectra of the reference substances is a good starting point for the EXAFS spectra fitting. As can be seen in Figure 5, the Zn $K$-edge spectrum of pure sphalerite in the R-range from 1.6 to 4.3 Å (not corrected for the phase shift) is accurately approximated by the three nearest-to-Zn coordination shells. The Fourier transform (FT) magnitude is characterized by the first intense peak, with a maximum at 1.8–2.0 Å, and two broad peaks located at R, from 3 Å to 4.5 Å. The results of the data analysis (Table 3) show that the first intense peak corresponds to the nearest 4 S atoms (Zn–S bond length is 2.34 Å). The next two peaks correspond to 12 Zn at 3.83 Å and 12 S at 4.49 Å. In contrast to pure ZnS, the FT magnitude of $CuInS_2$ and $AgInS_2$ EXAFS (In, Cu, and Ag $K$-edges) exhibits only one intense peak corresponding to the first coordination shell, which consists of 4 S atoms at 2.47 Å (In), 2.31 Å (Cu) and 2.54 Å (Ag). The FT features at higher distances, which correspond to the second and third coordination shells, are weak compared to ZnS, which can be explained by the splitting of the coordination shells and cationic vacancies in the crystal structure [28] (Figure 2 and Table 2 in this study).

### 3.3.3. In $K$-Edge

According to the analysis of XANES and WT EXAFS spectra, the local atomic geometry of In in sphalerite is close to that of Zn. This model implies that In substitutes for Zn in the sphalerite structure. Therefore, coordination numbers were adopted from the pure sphalerite structure and fixed during the fits. The nearest neighbors of the cation in the sphalerite structure are four S atoms; 12 Zn and 12 S atoms are located in the second and third coordination shells, respectively (Figure 2). A model with In atoms in the second coordination shell did not yield any significant improvement in the fit. Analysis of the experimental spectra demonstrated that the local atomic environment of In in all studied minerals is equivalent, regardless of the chemical composition (Table 2, Figure 7). In Sample 3757, where In is the only admixture, In has four S atoms in the first coordination shell

($R_{In-S}$ = 2.45 ± 0.01 Å), 12 Zn atoms in the second shell ($R_{In-Zn}$ = 3.91Å ± 0.01 Å), and 12 S atoms in the third shell ($R_{In-S}$ = 4.48Å ± 0.03 Å). All interatomic distances increased in comparison with pure sphalerite. The absence of In atom in the second coordination shell means that the In–In clustering is absent. The mentioned values of the interatomic distances are very close to the values obtained for Samples 4108, 4186, and 4197, where In presents in sphalerite along with other dopants, namely Cu and Ag. The In–S distances in Ag–In-bearing Sample 4169 exceed the distances in the other samples by 0.03–0.04 Å ($R_{In-S}$ = 2.49 and 4.54 Å in the first and third shell, correspondingly). The reason for this deviation is unknown.

As we did not observe any significant contribution of a heavy atom to the EXAFS signal in the distant coordination shells, In–In and In–Ag clustering is ruled out. The presence of Cu in the second coordination shell can not be detected, because EXAFS spectroscopy is not capable of distinguishing between contributions of Zn and Cu.

### 3.3.4. Cu *K*-Edge

The structural model of cubic sphalerite used to fit In *K*-edge EXAFS spectra does not provide good quality fits in the case of Cu *K*-edge. To accurately approximate the spectra of Samples 4108 and 4186, a distorted sphalerite structure was used. The first feature in the FT EXAFS curve corresponds to four S at ~2.30 Å. Fits of the FT features at 3–4.5 Å (not corrected for phase shift) resulted in the splitting of the second and third coordination shells of Cu, which consist of 12 Zn and 12 S atoms, respectively. The splitting of the distant coordination shells, caused by variation in the Cu–Zn and Cu–S interatomic distances, indicates to the significant distortion of ZnS crystal structure because of the isomorphic substitution Cu → Zn.

The spectrum of Sample 4065, which contains only Cu as a dopant, was accurately described by the model of covellite (CuS). Atoms of Cu in the structure of covellite occupy two inequivalent positions with triangular ($Cu_{D3h}$) and tetrahedral ($Cu_{Td}$) geometries (Tagirov et al. [17]). The EXAFS data analysis shows that $Cu_{D3h}$ is coordinated by 0.8 ± 0.2 S atoms at $R_{Cu-S}$ = 2.16 ± 0.02 Å and by two S atoms at $R_{Cu-S}$ = 3.63 ± 0.02 Å. The first coordination shell of $Cu_{Td}$ consists of 1.8 ± 0.2 S atoms at $R_{Cu-S}$ = 2.28 ± 0.02 Å and 0.6 S atoms at $R_{Cu-S}$ = 2.31 ± 0.02 Å. The third coordination shell is presented by two Cu at 3.10 ± 0.03 Å and the fourth shell of two S at 3.45 ± 0.02 Å. The interatomic distances in the first and second coordination shells are close to those of pure covellite but differ notably from the pure covellite structure in the third and fourth coordination shells. Our experimental data can be explained by the presence of the admixture of non-stoichiometric Cu (i) sulfides or the solid solution Cu (Car et al., [28]).

### 3.3.5. Ag *K*-Edge

Results of the Ag *K*-edge EXAFS spectra fitting show that the solid solution model best describes Sample 4197, with an Ag→Zn substitution. The contributions of scattering atoms are distinguishable up to the third coordination shell of Ag. The first coordination shell exhibits a considerable expansion of 0.12 Å compared to the pure sphalerite structure. The second coordination shell splits into two subshells and consists of 6+6 Zn atoms at 3.81 and 4.00 Å. The Ag–S distance of 4.48 Å in the third shell is close to the Zn–S distance in pure sphalerite (4.46 Å, Table 3). The high values of the Debye–Waller factors and the high uncertainty of the interatomic distances of the second and third coordination shells show that the local atomic structure around Ag is highly disordered. Despite the high uncertainty of the data obtained for the distant coordination shells of Ag, two reasons argue for the formation of the Ag-bearing solid solution. Firstly, the Ag–S distance in the first coordination shell of Ag is different from that of AgInS$_2$: 2.46 Å in Sample 4197 vs. 2.54 Å in AgInS$_2$ (Table 3). Secondly, the DFT calculations suggest that the first shell In–S and Ag–S distances in the sphalerite solid solution are equal (Table S3 of Supplementary Materials). Results of the EXAFS spectra fitting yield $R_{In-S}$ = 2.45 Å and $R_{Ag-S}$ = 2.46 Å for Sample 4197. Therefore, we believe that the main form of occurrence of Ag in Sample 4197 is the sphalerite solid solution.

For the Samples 4169 and 4152, good quality fits were obtained using the local atomic geometry of metallic Ag as the dominant form of silver. These results are consistent with the analysis of WT EXAFS. The LCF analysis of Ag *K*-edge XANES and EXAFS spectra (Table S2 of Supplementary Materials) showed the presence of $Ag_2S$ in Sample 4152. The content of $Ag_2S$ was determined as 18–19% from XANES and as 12% from EXAFS spectra LCFs. However, since the Ag–Ag scattering path intensity is much higher than that of Ag–S, the contribution of a light atom like S is masked by an FT feature of Ag at ~2 Å (right panel in Figure 8).

## 4. Implications

Sphalerite is the most important source of In. The concentration of this "critical" metal in natural minerals is usually strongly correlated with Cu content. In the present work, the state of In as a function of the chemical composition of the mineral was investigated. We synthesized sphalerite crystals which contained In as the only admixture, along with the samples in which In presented together with the 11th group elements Cu and Ag. Besides this, two samples were doped by one metal: Cu or Ag without In. The state of the admixtures, which occurred in the sphalerite crystals in the "invisible" form, was studied by means of X-ray absorption spectroscopy. Our experimental data allow us to discuss the chemistry of trace elements in sphalerite to elucidate the following issues: (i) how the state of In is affected by the chemical composition of sphalerite; (ii) how the state of 11th group metals in sphalerite changes with the increasing ionic radius of the element.

Results of the present study, combined with our previously obtained data for Au-bearing sphalerite (Filimonova et al., [1]), demonstrate that In can form a solid solution with sphalerite, independently of the presence of the other dopants. For the present study, we synthesized sphalerite, which contained 0.7 wt.% of In as the only admixture. In this sample, In presents in the solid solution state, substituting for Zn. Due to the incorporation of In, the local atomic environment of the cationic site where the substitution takes place is modified. The bond lengths increase (with respect to Zn in pure ZnS) from 2.34 to 2.45 Å in the first coordination shell ($N_S = 4$), from 3.85 to 3.91 Å in the second coordination shell ($N_{Zn} = 12$), and are close to the Me–S distance in the pure sphalerite in the third coordination shell ($N_S = 12$, $R_{In–S} = 4.48$ Å). Formation of the solid solution in the case of In-bearing sphalerite can take place via the charge compensation scheme $3Zn^{2+} \leftrightarrow 2In^{3+}+\square$, where $\square$ is a Zn vacancy.

In the presence of the 11th group metals (Cu, Ag, Au), the concentration of In in the solid solution state increases notably. Schorr et al. [4,5] found that Cu and In form a partial binary solid solution, $Zn_{2x}Cu_{1−x}In_{1−x}S_2$, with a miscibility gap in the region of $0.1 \leq x \leq 0.4$. In other words, the maximum concentration of In in the sphalerite solid solution of the composition $Zn_{0.8}Cu_{0.6}In_{0.6}S_2$ ($x = 0.4$) reaches 30 wt.%. We note here that the temperature of the miscibility gap was not reported, but these data suggest that the concentration of In and Cu in sphalerite can be very high in cases in which these elements present together. Our data, and the results of the previous studies [1,21], demonstrate that all the 11th group metals can form isomorphous solid solution at a high temperature, according to the scheme $2Zn^{2+} \leftrightarrow Me^+ + In^{3+}$ (Me = Cu, Ag, Au). This scheme implies that the increase in the concentration (solubility) of both elements—$Me^+$ and $In^{3+}$—in the sphalerite solid solution is directly correlated with the temperature. Upon cooling from the ore formation temperature to the ambient conditions, the stability fields of solid solutions contract, which can result in the exsolution of the solid solution components. Our data demonstrate that at ambient T-P conditions, In and Cu are retained in the solid solution state. In the solid solution, Cu occupies a cationic position in the ZnS structure. The first coordination shell of In slightly contracts compared to the pure sphalerite ($R_{Cu–S} = 2.30–2.31$ Å vs $R_{Zn–S} = 2.34$ Å). The second shell composed of 12 Zn atoms is split into two groups of Zn atoms ($R_{Cu–Zn} = 3.76–3.8$ Å and 3.92–4.0 Å). The third shell consists of 12 S atoms and is also distorted in comparison with pure sphalerite, but the average Cu–S distance only slightly (by 0.01–0.05 Å) exceeds the Zn–S distance in pure ZnS. Our data suggest that Ag, like Cu, can form a solid solution with In-bearing sphalerite at high temperatures. After cooling to ambient temperature, Ag is retained in the

form of a solid solution. However, in contrast to the Cu-bearing sphalerite, the distant coordination shells of Ag are more distorted because of a significant difference in the ionic radii with Zn.

Combining the results of the present study with the data of Filimonova et al. [1], it is possible to follow the state of the 11th group metals from Cu to Au. At high temperatures, all the metals can form a solid solution with sphalerite via the charge compensation scheme $2Zn^{2+} = Me^+ + Me^{3+}$. Apart from $In^{3+}$, it can be a cation of another chalcophile element—for example, $Tl^{3+}$, $As^{3+}$, $Sb^{3+}$, $Bi^{3+}$, or $Fe^{3+}$. In the study of Tonkacheev et al. [21], we found that the concentration of Au in sphalerite is correlated with the Fe content, which means that the substitution scheme given above is realized because of the partial oxidation of Fe to the 3+ state. At ambient temperature, the state of the $11^{th}$ group elements in sphalerite depends on the ionic radius, which increases in the order of Cu (0.6 Å), Ag (1 Å), Au (1.37 Å for CN = 6) [29]. Copper withstands cooling and at ambient temperature exists in the solid solution state as its ionic radius is equivalent to that of Zn (0.6 Å). Silver is also present in the solid solution form at ambient temperature, but its first shell exhibits significant expansion in comparison with the pure ZnS, whereas the second and third coordination shells are split into two subshells each. Gold mostly exsolves from the solid solution with the formation of $Au_2S$-like clusters [1]. Only traces of the solid solution Au were detected in measurements performed at ambient temperature. The Me–S distances in the first coordination shell in the sphalerite solid solution are correlated with the ionic radii and increase as 2.30–2.31 (Cu) < 2.46 (Ag) < 2.5 (Au).

The results of our study, as well as data from Filimonova et al. [1], demonstrate that, despite the charge compensation substitution scheme, $Me^+$–$In^{3+}$ clustering is absent. We did not detect any notable contribution from a heavy atom to the EXAFS signal corresponding to the second coordination shell of In, Cu, and Ag. Moreover, according to the results of the DFT simulations (Table S3 of Supplementary Materials), the $Me^+$–$In^{3+}$ clustering would produce a notable splitting of interatomic distances in the second coordination shell, which was observed in the results of the EXAFS spectra fitting in the case of Cu and Ag. However, the applied conventional EXAFS fitting procedure is not able to provide reliable structural information in the case of the disordered crystal structures around impurities.

**Supplementary Materials:** The following are available online at http://www.mdpi.com/2075-163X/10/7/640/s1, Table S1: Position of absorption edge and white line of In *K*-edge, Cu *K*-edge, and Ag *K*-edge XANES spectra; Table S2: Results of Linear combination fit analysis of Ag *K*-edge XANES and EXAFS spectra; Table S3: Interatomic distances in sphalerite determined by DFT calculations, Figure S1: Results of linear combination fit analysis of Ag *K*-edge XANES and EXAFS spectra.

**Author Contributions:** N.D.T. participated in the XAS experiment, performed the XAS data treatment and interpretation, and performed chemical analyses; A.L.T. coordinated the XAS data treatment and interpretation, XRD analyses, and performed DFT calculations; B.R.T. designed the study, participated in the XAS experiment, and wrote the paper; O.N.F. participated in the XAS experiment and data treatment; P.V.E. and D.A.C. carried out the synthesis experiments; K.O.K. organized the XAS experiment at BM 20 (ESRF); M.S.N. participated in the XAS experiment and performed SEM and XRD analyses; all the authors participated in the manuscript preparation. All authors have read and agreed to the published version of the manuscript.

**Funding:** This study was supported by the Russian Science Foundation, grant No. 18-77-00078 (XAS experiments, DFT calculations, experimental data treatment). A.L.T. and K.O.K. acknowledge support by the Russian Ministry of Science and Education under grant No. 075-15-2019-1891 (XAS experimental set-up and technique design).

**Acknowledgments:** The authors thank the ESRF scientific council for the beamtime allocation under proposal ES-703 (ROBL). The help and support of Andre Rossberg and Nils Baumann (ROBL) is greatly appreciated. We are grateful to Vera Abramova for the chemical analyses of synthesized minerals using the LA-ICP-MS methods and to Elena Kovalchuk for the EPMA analyses. Quantum chemical calculations have been carried out using computing resources of the Federal Collective Usage Center, Complex for Simulation and Data Processing for Mega-Science Facilities at NRC "Kurchatov Institute" (ministry subvention under agreement RFMEFI62117X0016).Chemical analyses were performed at the "IGEM-Analytica" Center for Collective Use.

**Conflicts of Interest:** The authors declare no conflict of interest.

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
