# Peer review of "The State of Trace Elements (In, Cu, Ag) in Sphalerite Studied by X-Ray Absorption Spectroscopy of Synthetic Minerals"

_minerals, doi:10.3390/min10070640_

Round 1

Reviewer 1 Report

Please, see my annotated commnents in the ms

Author Response

Dear Reviewer,

thank you for reading our manuscript. We have accepted almost all suggesetdions to improve the text of our manuscript.

Size of the Inclusions part. We would like to keep quite big "Inclusions" part since our manuscript summarizes our efforts in the investigation of the state of In and conjugated metals in sphalerite. In the "Implications" section we not simply summarize results of the present study, but aply the results to analysis of the  behaviour of the admixtures in sphalerites of various compositions which formed in different natural environments. We believe that this section will attract attention of scientists who are not specialized in XAS spectroscopy but are interested in shpalerite chemistry, and mineralogy and crystal chemistry in general (because our study explains some general principles of isomorphous substitutions in natural minerals). The "Implications" section is commonly accepted in many scientific journals such as American Mineralogist, Chemical Geology, and so on.  Therefore, we would prefer to keep this section as it is.

Concearning the concentrations of dopant elements:

The number of synthesized crystals was much greater than we studied by XAS, and the composition of the initial compositios satisfied the comment of the Reviewer. But not all of the synthesized crystals were suitable for XAS experiment because of contamination by other phases or problems with transport of the initial reagents, when no sphalerite crystals were observed at the cold end of the ampoule. For XAS experiment we selected the best crystals of contrasting compositions.

Line 344: "The reason of this deviation is unknown". Answer: 

For this sample we also performed EXAFS spectra fittings with In-S interatomic distances fixed to the values obtained for other samples. In this case the fit quality decreased substantially. The XRD pattern of this sample corresponds to pure sphalerite with "normal" cell parameters, we double checked this. Therefore, the observed deviation is an experimental result which we, at the moment, cannot explain. We agree with the Reviewer that it would be nice to explain this result, but, as we do not have an idea about the reason of this deviation, we would prefer to avoid speculations as they can be misleading.

Reviewer 2 Report

This MS (# Minerals-860876) deals with the characterization of the oxidation state and local environment of synthetic sphalerite crystals doped with In, Cu and Ag. The investigation has been carried out by means of X-ray absorption spectroscopy above the In, Ag, and Cu K-edge coupled with the results of DFT simulations. Scanning electron microscopy, X-ray powder diffraction, electron probe micro-analysis (EPMA) and laser ablation inductively coupled mass spectrometry were also used to characterize phase compositions of synthetized samples.

The MS is properly arranged, well-written and informative, the structural and chemical characterization has been properly assessed. In conclusion, this work is of interest for the scientific community and worth to be published in the MDPI journal Minerals after minor revision suggested as follows.

-Experimental Details of x-ray diffraction analysis, as well as details about DFT calculations, are lacking in the text. A mention of a reference, concerning only the DFT method, is provided in Table S3.

-the Authors claim that, on XRD results basis (lines 177-181), all samples composition corresponds to the only sphalerite phase.  On the other hand, SEM/EDS analyses (lines 184-186) detected two phases for sample 4197. The last statement is reiterated in XANES spectra analysis section (lines 251-253). The content of Ag2S was also estimated as 18-19% from XANES and as 12% from EXAFS spectra. These phase concentrations are easily handled by X-ray powder diffraction. The same occurs for sample 4065 with covellite CuS (lines 238-241, 276-277 and 364-365). How do the Authors explain these contradictory statements with respect to XRD results?

-As concerns the spectra fit of samples 4108 and 4186 the Authors should better describe the distorted model used.

-Last section is entitled “Implications”, but “Conclusions” is more appropriate.

Lines 90, 97, 101,104: change “charge composition” with “starting composition”

Line 120: change “lines and crystals used…” with “lines and diffracting crystals”

Line 195: change “Coordination polyhedra around..” with “1st, 2nd, and 3rd shell coordination polyhedra around….”

Lines 196, 197, 202: please provide the right reference. Schorr et al (2006) performed a phase transition study. They did not give CuInS2 structural details.

Line 206: change “indicate position of W.L. for In2S3” with “indicate position of WL for In2S3

Line 307:  change “Table 2. Structural parameters” with “Table 3. Structural parameters”

Line 309: same comment as above (line 202)

Line 336: change “(Table 2, Figure 7).” with “(Table 3, Figure 6).”

Line 439: change “coordination shell of In slightly” with “coordination shell of Cu slightly”

Author Response

Dear Reviewer,

thank for the reading our manuscript and suggested improvements.

Experimental Details of x-ray diffraction analysis, as well as details about DFT calculations, are lacking in the text. A mention of a reference, concerning only the DFT method, is provided in Table S3.  

We agree with these suggestions and details of the X-ray diffraction experiments and DFT calculations have been added to the manuscript.

the Authors claim that, on XRD results basis (lines 177-181), all samples composition corresponds to the only sphalerite phase.  On the other hand, SEM/EDS analyses (lines 184-186) detected two phases for sample 4197. The last statement is reiterated in XANES spectra analysis section (lines 251-253). The content of Ag2S was also estimated as 18-19% from XANES and as 12% from EXAFS spectra. These phase concentrations are easily handled by X-ray powder diffraction. The same occurs for sample 4065 with covellite CuS (lines 238-241, 276-277 and 364-365). How do the Authors explain these contradictory statements with respect to XRD results?

Table 1 presents concentrations of Ag and Cu in synthesized samples. For samples 4152 and 4169 the concentration of Ag is lower than 0.1 wt. %, which is almost undetectable by X-ray diffraction. Concentrations of Ag2S determined by XANES or EXAFS techniques point out the relative concentration of Ag2S to the whole amount only of Ag, not to the entire sphalerite sample. For example, concentration of Ag in sample 4152 is 23 ppm, only 18 % of Ag is in Ag2S phase, therefore concentration of Ag2S in the sample is 23 ppm*0.18= 4.1 ppm. This level is surely undetectable by X-ray diffraction. These statements concern also other imputities in the sphalerite samples.

As concerns the spectra fit of samples 4108 and 4186 the Authors should better describe the distorted model used.

The disordered model used to fit EXAFS spectra for samples 4108 and 4186 consists of the splitting of the second and the third coordination shell near the absorbing atoms. But the used technique doesn't provide reliable structural parameters for disordered atomic structures since the obtained result depends on the starting model. The aim of such fitting consists of in demonstration of considerable geometrical distortions near embedded atoms. Unreliability of conventional EXAFS procedure applied for disordered atomic systems is pointed out in section Inclusions. 

Last section is entitled “Implications”, but “Conclusions” is more appropriate.

Our manuscript summarizes our efforts in the investigation of the state of In and conjugated metals in sphalerite. In the "Implications" section we not simply summarize results of the present study, but aply the results to analysis of the  behaviour of the admixtures in sphalerites of various compositions which formed in different natural environments. We believe that this section will attract attention of scientists who are not specialized in XAS spectroscopy but are interested in shpalerite chemistry, and mineralogy and crystal chemistry in general (because our study explains some general principles of isomorphous substitutions in natural minerals). The "Implications" section is commonly accepted in many scientific journals such as American Mineralogist, Chemical Geology, and so on.  Therefore, we would prefer to keep this section as it is.
